# Influence of B and Nb Additions and Heat Treatments on the Mechanical Properties of Cu-Ni-Co-Cr-Si Alloy for High Pressure Die Casting Applications

**Denis Ariel Avila-Salgado** [1,*], **Arturo Juárez-Hernández** [1,*], **Fermín Medina-Ortíz** [1],
**María Lara Banda** [2]  **and Marco Antonio Loudovic Hernández-Rodríguez** [1]

[1] Facultad de Ingeniería Mecánica y Eléctrica (FIME), Universidad Autónoma de Nuevo León,
Av. Universidad S/N, San Nicolás de los Garza, Nuevo León 66450, Mexico;
fermin.medinarz@uanl.edu.mx (F.M.-O.); marco.hernandezrd@uanl.edu.mx (M.A.L.H.-R.)

[2] Centro de Investigación e Innovación en Ingeniería Aeronáutica (CIIIA), Facultad de Ingeniería Mecánica y
Eléctrica, Universidad Autónoma de Nuevo León, Carretera a Salinas Victoria km. 2.3, Apodaca C.P.,
Nuevo León 66600, Mexico; maria.laraba@uanl.edu.mx

* Correspondence: denis.uanl.mx@hotmail.com (D.A.A.-S.); arturo.juarezhn@uanl.edu.mx (A.J.H.);
Tel.: +52-812-571-1656 (D.A.A.-S.); +52-811-610-7184 (A.J.-H.)

**Abstract:** During the high pressure die casting process (HPDC), it is necessary to develop new designs and alloys for the copper plungers. In this research, two alloys Cu-9Ni-1Co-1.6Cr-2Si-1.3Fe-0.25B wt.% (A1) and Cu-9Ni-1Co-1.6Cr-2Si-0.1Fe-0.2Nb wt.% (A2) under different heat treatments (HT) were studied. Optical microscopy technique was applied to reveal the regions of dendritic morphology, also lower Secondary Dendrite Arm Spacing (SDAS); and different grain orientations. The results reveal that the solidification sequence is primary Cu dendrites and secondary intermetallics; heat treatments increase the redistribution of alloying elements in the interdendritic regions. During the heat treatments, some precipitates were found in the grain boundary after aging heat treatments for both alloys, which were determined by X-ray diffraction. Hardness test HRB presented a decrease with the solution heat treatment and an increase with the aging heat treatments proposed for both alloys. Finally, the wear resistances for both alloys were compared with a commercial alloy C17530, with decreased A1 with B additions having the best result in the as-cast condition $4.07 \times 10^{-4}$ mm$^3$/Nm, while for A2 with Nb additions wear resistance increased, with the best result in the one with aging heat treatment $1.69 \times 10^{-4}$ mm$^3$/Nm while for the C17530 alloy this was $2.74 \times 10^{-4}$ mm$^3$/Nm.

**Keywords:** Cu-Ni-Co-Cr-Si alloy; grain refiners; heat treatments; microstructure; dendritic coherence; precipitation hardening; wear on copper alloys

## 1. Introduction

Copper increases its performance, such as electrical resistance, when it is alloyed with different elements [1]; alloys such as Cu-Ni-Si have been used in electrical applications due to their excellent electrical conductivity and mechanical stress properties [2]. Recently, Cu-Ni-Si-Cr alloys have been improved by changing the Ni/Si relation and optimizing the heat treatments in order to improve their mechanical properties [3]. Mechanical properties for Cu-Ni-Si alloys can be improved by additions of other elements; Al [4] and Mg [5] improve solidification kinetics and intermetallic precipitations; Ti can refine the microstructure due to the reduction of solubility of Ni and Si in the copper matrix [6]; Cr can refine the structure by precipitation of Cr$_3$Si particles [7]; Zr [8] and V [9] can modify the microstructure and improve the performance of aging heat treatment; Co [10] additions affect microstructure and increase hardness by aging heat treatment. Cu-Be are the most used alloys as plungers in the HPDC

industry such as Cu-Be (C17510 and C17530), but these alloys have a problem due to the Be toxicity during the casting process, which produces inorganic beryllium exposure and tends to occur via inhalation of beryllium fumes or dust, but it can also be absorbed following skin exposure, causing a serious health disease called berylliosis [11–15]; this is not in the casing process but during the melting of Be and Be master alloy manipulation. On the other hand, aging heat treatment is used to increase tensile strength and hardness during 1 or 5 hours depending of the alloy. Having the melting temperature as a reference, $\frac{1}{2}$ or $\frac{3}{4}$ of this temperature is used as the heat treatment temperature. Aging heat treatment is made by a previous solid solution heat treatment after casting the alloy; the more elements in solution, the higher the hardness and mechanical resistance [16].

A thermodynamic study made by Meijering [17] in a Cu-Ni alloy shows that it could be an immiscibility region under a certain alloy composition range but experimentally it has not been observed, this shows that the Cu-Ni system is not suitable for heat treatments. This could be modified by additions of a third element such as Cr, Si, Co, and so on which make these alloys harden by precipitation through heat treatment [18,19]. Within the development of copper alloys with high mechanical properties, some authors have studied the addition of new elements in the copper matrix. Klement et al. [20] in the patent U.S. 3,072,508 proposed a copper alloy free of Be, Ni (0.75–2%), Cr (0.25–1.25%), Si (0.25–1%) and Cu balance in wt.% and found higher wear resistance and hardness employing some heat treatments proposed by Gorson [21], where they found hardness values of 75–85 HRB. Walter et al. [22] in the patent U.S. 4260435 proposed another alloy free of Be; this alloy is Ni (2–3%), Co (0.4–0.8%), Si (0.1–0.5%), Cr (0.1–0.5%) and Cu balance in wt.% and it is treated by a solution heat treatment, quenched in air and followed by two aging heat treatments in order to form some precipitates and increase the hardness; the second aging heat treatment was applied due to the low solubility of Cr in Cu.

Due to these reasons, this research is a continuation of a previous work on Cu-9Ni-1Co-1.6Cr -2Si [23], which gave the best results of 85.7 HRB and a wear rate of $2.1 \times 10^{-4}$ mm$^3$/Nm. It is proposed that adding other alloying elements gives the same mechanical properties or improves them, so Cu-9Ni-1Co-1.6Cr-2Si alloy with additions of B and Nb as grain refiners is proposed; percentages were chosen based on their maximum solubility from the respective binary Cu phase diagrams. Microstructural evolution was studied using optical and electronic microscopies and X-ray diffraction techniques, hardness and wear tests were made in order to understand and analyze the phenomena involved such as friction, lubrication and wear [24], microstructural evolution with these properties, and the influence off B and Nb in base Cu-Si-Co-Cr-Ni alloys.

## 2. Materials and Methods

### 2.1. Sample Preparation

The material used in the present work was made of Cu, Si, Co, Cr, and Ni 99.99 wt.% purity and Nb65-Fe32-Si3 and B18-Fe82 alloys in wt.% were melted at 1300 °C in an induction furnace INDUTHERM TF4000 (Induterm, Walzbachtal, Germany) in a SiC crucible and degassed with N$_2$ for 15 min. Liquid metal was cleaned with a cover flux and casted in a cylindrical permanent mold steel H13, with a height of 12 cm and a diameter of 2.54 cm. The chemical composition of the samples was made by spark emission spectrometer SPECTRO model LAB (Analytical Instruments, Kleve, Germany); Table 1 shows the results of chemical composition analysis of the two samples, A1 and A2 alloys. Samples were machined up to 1 cm in order to keep a homogenous grain distribution in the samples for the next experimental techniques. Optical and electronic microscopy, X-ray diffraction techniques and hardness and wear tests were conducted in order to understand microstructural evolution with these properties, and the influence of B with Fe in order to form precipitates, and Nb on the Cu-Ni-Co-Cr-Si base alloys.

**Table 1.** Alloy chemical compositions in wt.%.

| Alloy | Ni | Co | Cr | Si | Fe | B | Nb | Cu |
|-------|----|----|-----|----|-----|------|------|---------|
| A1 | 9 | 1 | 1.6 | 2 | 1.3 | 0.25 | - | Balance |
| A2 | 9 | 1 | 1.6 | 2 | 0.1 | - | 0.20 | Balance |

## 2.2. Heat Treatments

Casting bar sections were cut, and these sections were given different heat treatments in an electric furnace as proposed by [25,26], so the first sample was as cast (AC); then a solution heat treatment (TS), an aging heat treatment (TA1), and finally a second aging heat treatment (TA2) were given. Time and temperatures of the different heat treatments are shown in Table 2.

**Table 2.** Design of the experiment and identification of samples.

| Sample | AC/HT | Temperature (°C) | Time (Min) | Analysis Condition |
|--------|-------|------------------|------------|--------------------|
| S1 | AC | 1300 | - | AC |
| S2 | TS | 925 | 120 | AC + TS |
| S3 | TA1 | 450 | 100 | AC + TS+ TA1 |
| S4 | TA2 | 550 | 180 | AC + TS + TA1 + TA2 |

AC = as-cast, TS = solution heat treatment, TA = aging heat treatment.

## 2.3. Sample Characterization

Sample characterization was made for all samples, namely S1, S2, S3 and S4, microstructure, X-Ray diffraction, hardness and wear tests. Specimens for metallography were prepared by a traditional method of grinding with abrasive paper followed by polishing with 1, 0.5 and 0.3 μm diamond paste. Afterwards, these samples were etched between 5 and 15 s by a solution made of 100 mL ethanol, 17 mm chloridric acid and 3.3 g ferric chloride, and washed with water at room temperature [27]. Samples were analyzed by optical and scanning electron microscopy, Jeol Model JSM 6510LV (Jeol Ltd., Akishima, Tokyo, Japan). X-Ray diffraction analysis was used in order to identify presented phases, Empyrean with monochromatic radiation CuKα. Grain size and Secondary Dendrite Arm Spacing (SDAS) were measured for each sample by the linear intersection method.

Rockwell B hardness measurements were made in an INCOR equipment (Mitutoyo corporation Ltd, Kawasaki, Kanagawa, Japan), with a 100-kg load and a ball of 1/16 in radius for 7 s. Each sample was tested ten times according to standard ASTM E18 [28]. The wear tests were conducted by a pin-on-disk tribometer (UANL, Monterrey, Nuevo León, México) according to ASTM G99-95a [29], on the surface previously polished on the specimens with dry sliding conditions; tests lasted 45 min with a distance of 539.5 m at constant velocity of 0.2 m/s, and an applied deadweight load of 30 N; the material of the pin was a H13 steel ball with a diameter of 11 mm. The mass loss was obtained by a geometric method computing the data measured in a coordinate microscopy according to:

$$\text{Disk volume loss} = V_1 = 2\pi R \left[ r^2 \sin^{-1}\left(\frac{d}{2r}\right) - \left(\frac{d}{4}\right)\left(4r^2 - d^2\right)^{1/2} \right] \tag{1}$$

where: $R$ = wear track radius, mm; $d$ = wear track width, mm; and $r$ = ball end radius, mm. Assuming that there is no significant wear on the ball.

The wear rate $K$ ($mm^3$/Nm) of the alloys was determined from the detached material's volume, by the following relationship:

$$K = \frac{V_1}{N \cdot S} \tag{2}$$

where: $V_1$ ($mm^3$) is the volume of material loss, $N$ (N) the normal load, and $S$ (m) the sliding distance.

## 3. Results and Discussion

### 3.1. Macrostructure and Microstructure Analysis

Macro and microscopic images are shown for samples S1, S2, S3 and S4, with both alloys with different crystallographic orientations having two regions with equiaxial grains in the center of the sample, and columnar grains near the walls of the samples. From Figure 1 for A1S1 and Figure 2 for A2S1 alloys, Nb and B play an important role in the kinetics of solidifications as they promote a higher density of nucleation sites; the columnar region is growing to the center until it reaches the equiaxial regions which nucleate in the liquid in the center of the sample. The transition from columnar to equiaxed occurs near the wall of the H13 ingot, which means that columnar grains have a crystallographic orientation aligned with the heat flow, while equiaxial grains have a miss-orientation with this heat flow; see Figure 1 for A1S1 and Figure 2 for A2S1 alloys. For both alloys, morphology is characterized as dendritic; see Figure 3 for A1S1 and Figure 4 for A2S1 alloys.

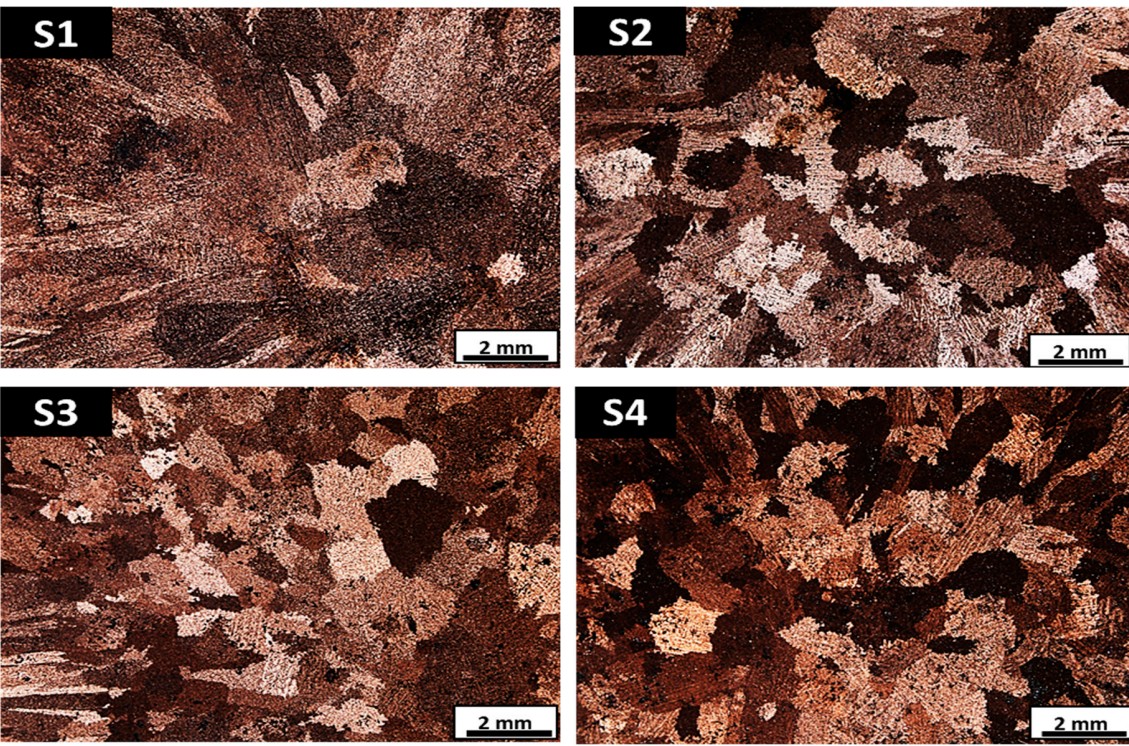

**Figure 1.** Stereoscope macrostructure for A1 alloy with four heat treatments: samples S1, S2, S3, and S4.

Another role that is played by Nb and B in the crystal is that Nb can be a substitutional atom while B can be an interstitial. Considering the value of the interstitial radius of Cu is 0.54 Å, this can be explained using the Hume–Rothery rules as the interstitial solubility of B on Cu, as the atomic radius is 23.3% smaller; even though it is unambiguous whether the ground state of boron in bulk Cu is interstitial or substitutional, it is reported that the B interstitial is more stable [30]. On the other hand, Nb solubility is substitutional and the atomic radius is 14.2% bigger; this situation promotes a delay in the recrystallization process increasing mechanical properties, and with the aging heat treatment processes increasing hardness due to the precipitation of secondary intermetallics acting as barriers and stresses for dislocations. Results for the different heat treatments show that grain size decreases from 1500 to 900 μm for A1 alloy, as shown in Figure 5 and from 1700 to 1100 μm for A2 alloy, as shown in Figure 6; this is the result of the solution heat treatment and quenching in water. Eventually, after heat treatments, S3 and S4 do not seem to change in grain size. SDAS seems to be unaffected, and some segregation is observed, as is shown in their microstructures for A1 alloy and for A2 alloy.

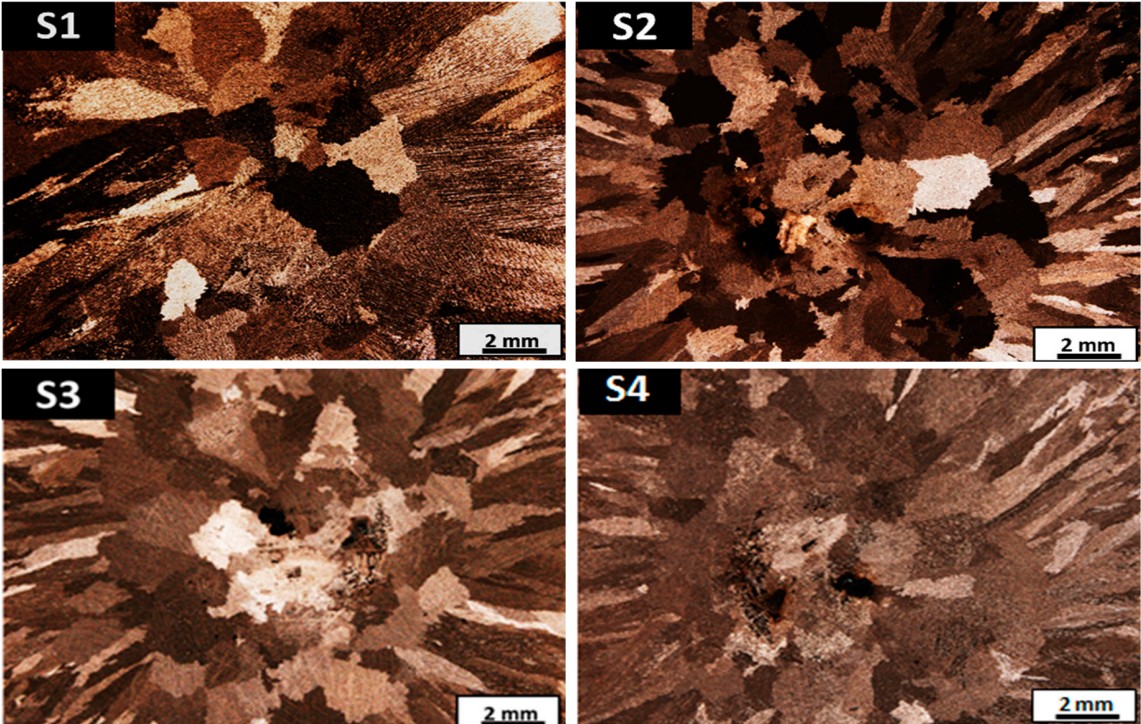

**Figure 2.** Stereoscope macrostructure for A2 alloy with four heat treatments: samples S1, S2, S3, and S4.

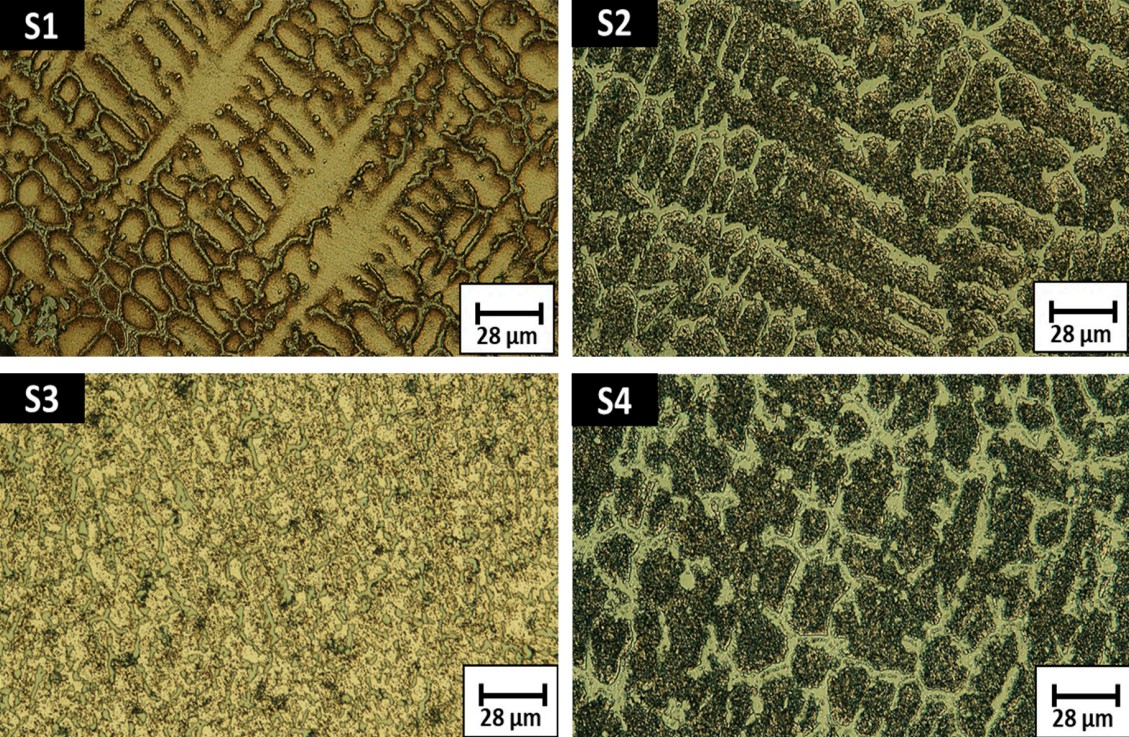

**Figure 3.** Optical microstructure for A1 alloy with four heat treatments: samples S1, S2, S3, and S4.

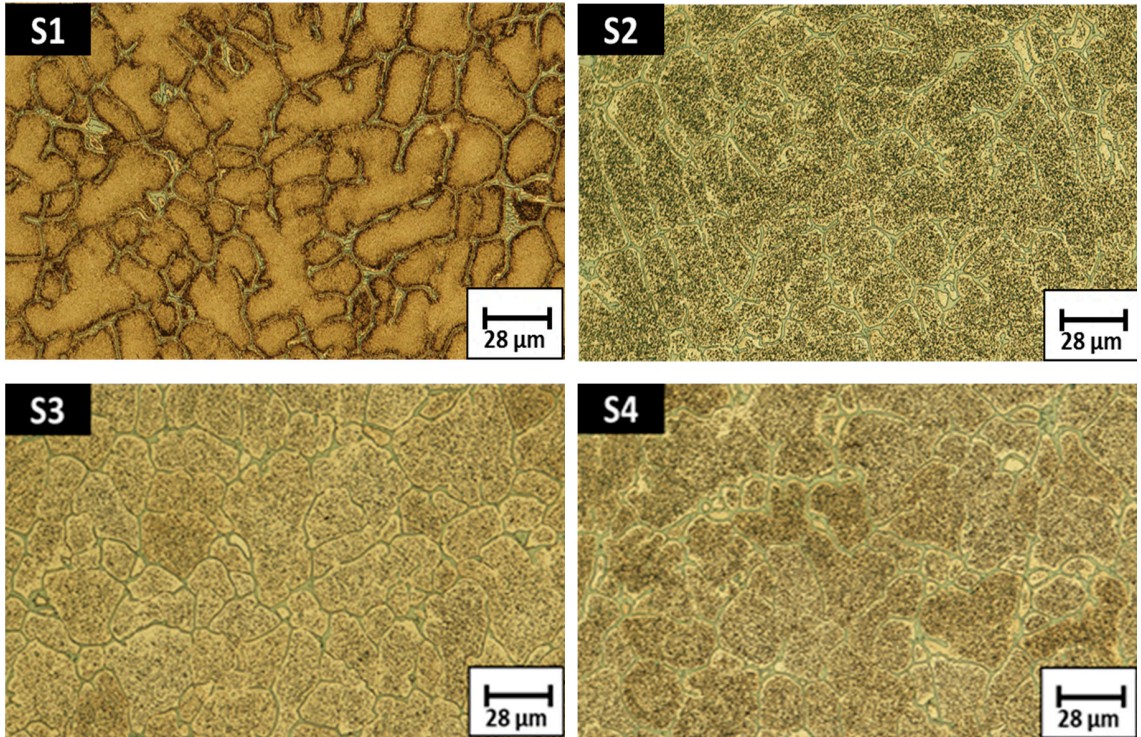

**Figure 4.** Optical microstructure for A2 alloy with four heat treatments: samples S1, S2, S3, and S4.

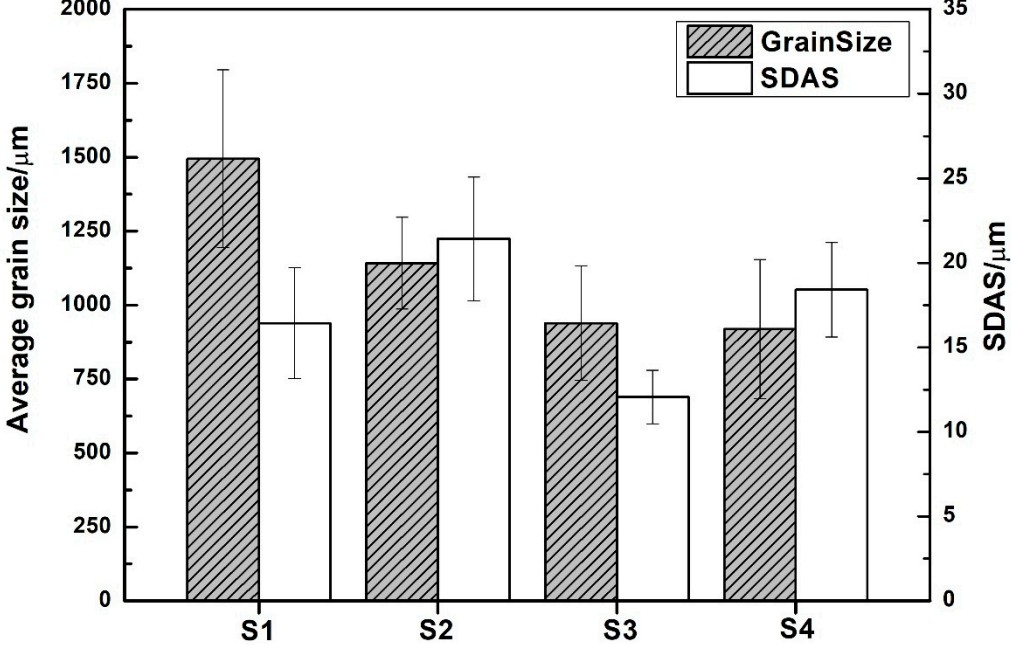

**Figure 5.** Average grain size and SDAS for A1 alloy with four heat treatments: samples S1, S2, S3, and S4.

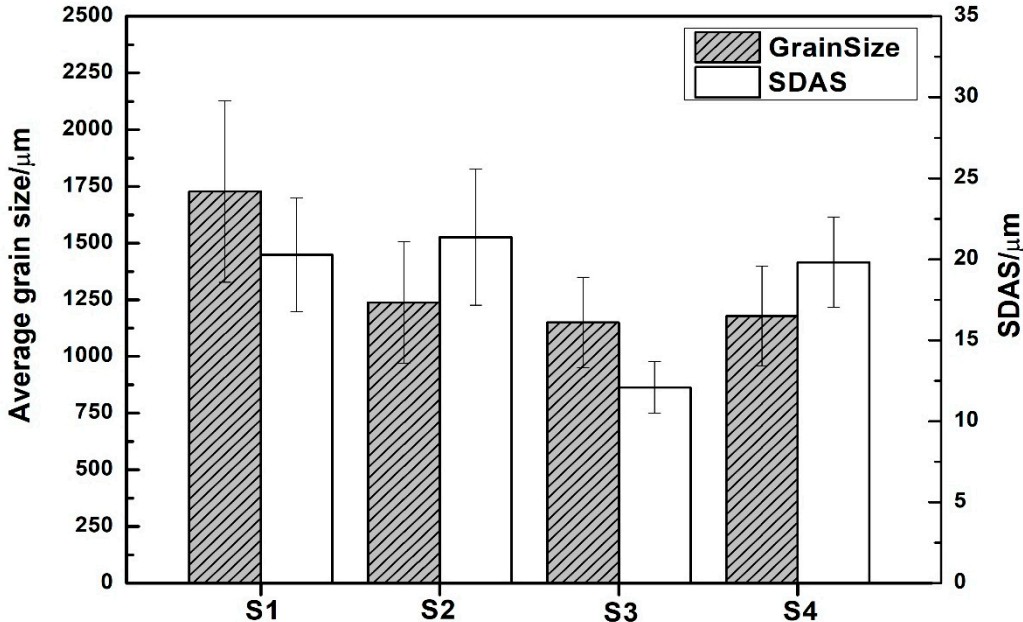

**Figure 6.** Average grain size and SDAS for A2 alloy with four heat treatments: samples S1, S2, S3, and S4.

### 3.2. Phase Analysis (XRD)

Figure 7 shows XRD patterns before and after heat treatments for A1 and A2 alloys. It is observed that the diffraction pattern peaks change in height for the copper matrix, showing preferences towards crystal orientations as temperature, time, and heat treatments increase. A high intensity of the peaks for both alloys is shown for the as-cast samples S1, which is the result of a high crystallographic orientation. Intensity decreased, showing that heat treatments change crystal sizes. It was also observed that the diffraction angle for Cu was moved as crystal parameters were modified due to the impurities in the matrix with the solution heat treatments. Some precipitates were found in the grain boundary after aging heat treatments; B1Cr1, CoNi, B1.43Ni4.29Si2 and Fe0.916Ni0.084 for A1 alloy peaks increase with heat treatments, while CrSi2 and Cu3Fe17 dissolve with the solution heat treatment, and CoNi and Nb6Ni16Si7 increase with S2, S3 and S4 heat treatments applied to A2 alloy.

### 3.3. Hardness Test

Figure 8 shows hardness test values, HRB, for the different heat treatments for A1 and A2 alloys. In both cases, there is a decrease as a result of the solution of the elements into the Cu matrix, decreasing from a value of 98.3 ± 0.70 for A1S1 to 76.10 ± 3.59 for A1S2, and from 91.9 ± 1.1 for A2S1 to 65.82 ± 4.37 for A2S2. A1S3 and A2S3 present the highest values of 101 ± 0.4 HRB as some intermetallics precipitate in the interdendritic zones, as shown by the XRD results and a distribution of the Co and Cr in the matrix. Finally, A1S4 and A2S4 present a small decrement in the hardness values, 98 ± 1.1 and 99 ± 0.5 HRB respectively, so it is considered that the second aging heat treatment decreases the properties and is not useful. Commercial C17530 alloy had 84.61 ± 2 HRB.

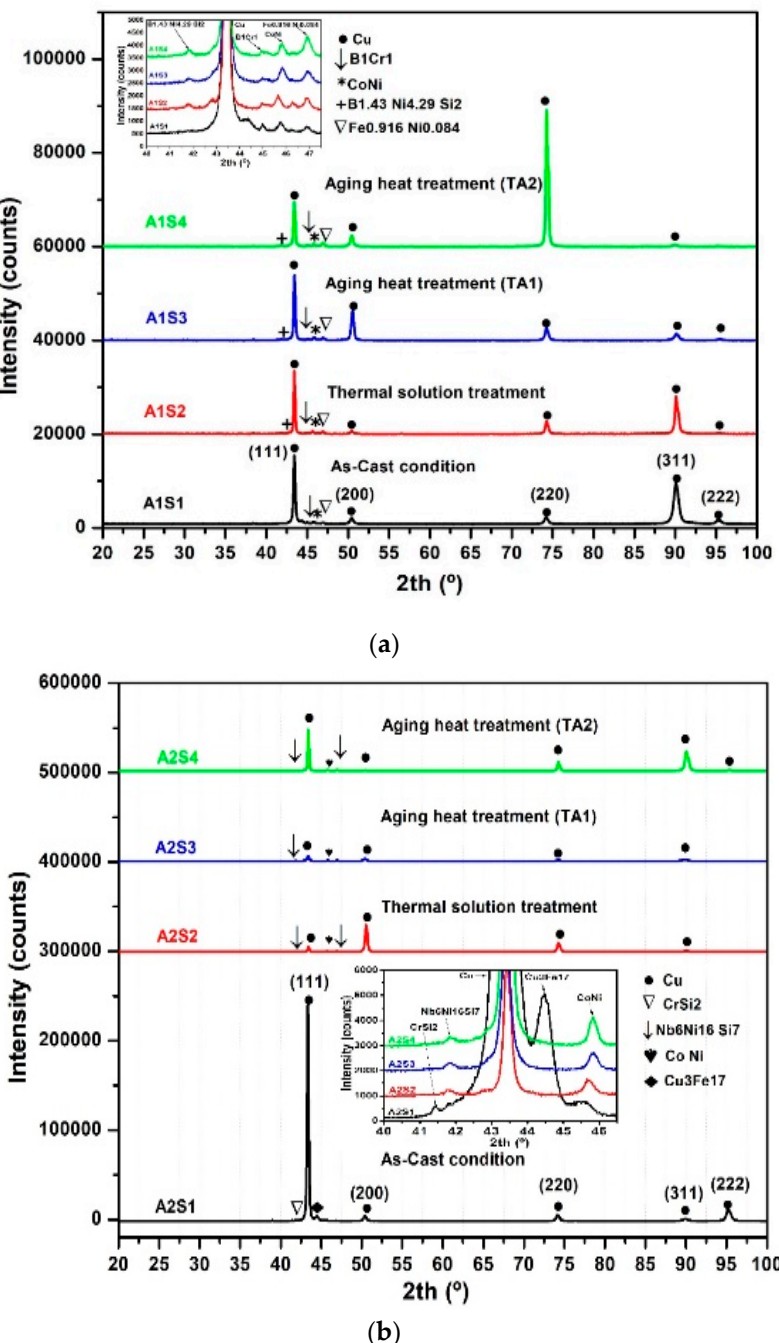

**Figure 7.** XRD results of alloys for the different conditions: (**a**) A1 alloy and (**b**) A2 alloy.

### 3.4. Analysis of Volume Loss and Friction on the Pin-On-Disk Machine

Figure 9 shows SEM micrographs of specimens A1S3, A2S3 and a commercial alloy C17530 that were carried out on a pin-on-disk tribometer. It can be noted that the wear mechanism is the same in both alloys; however, it was higher for the A1 alloy as it presented higher hardness, due to the precipitates formed having a higher deformation. For A1, wear resistance was decreased as volume loss was increased with the heat treatments, S3 and S4; the best wear condition for this alloy was the A1S1 condition ($4.07 \times 10^{-4}$ mm$^3$/Nm); see Figure 10. The best results were for the condition A2S3 ($1.69 \times 10^{-4}$ mm$^3$/Nm); this is explained as there is an initial grain refinement, A2S2 condition; then Nb is distributed in the copper matrix, so it is considered that this condition could be suitable for a plunger tip.

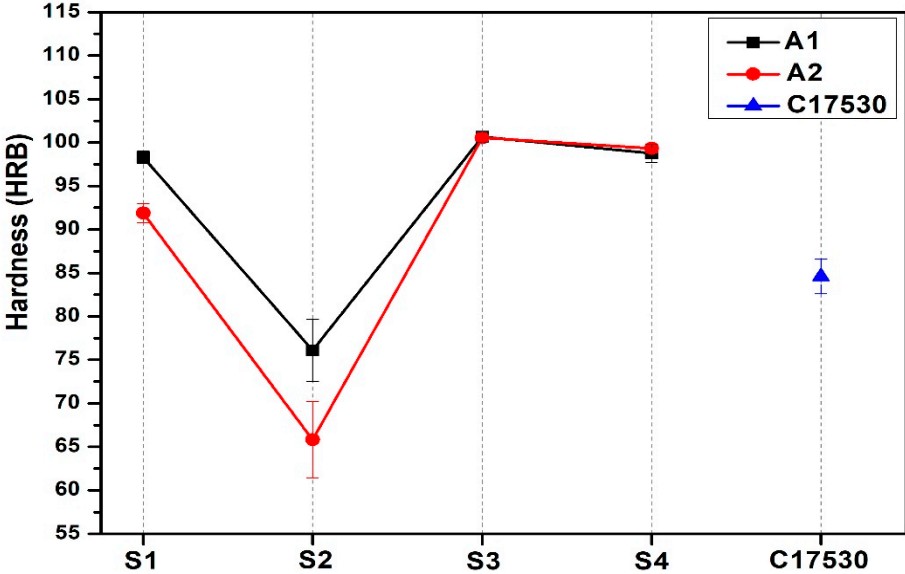

**Figure 8.** Rockwell B hardness for A1 and A2 alloys with different heat treatments on the samples S1, S2, S3 and S4 and C17530 alloy.

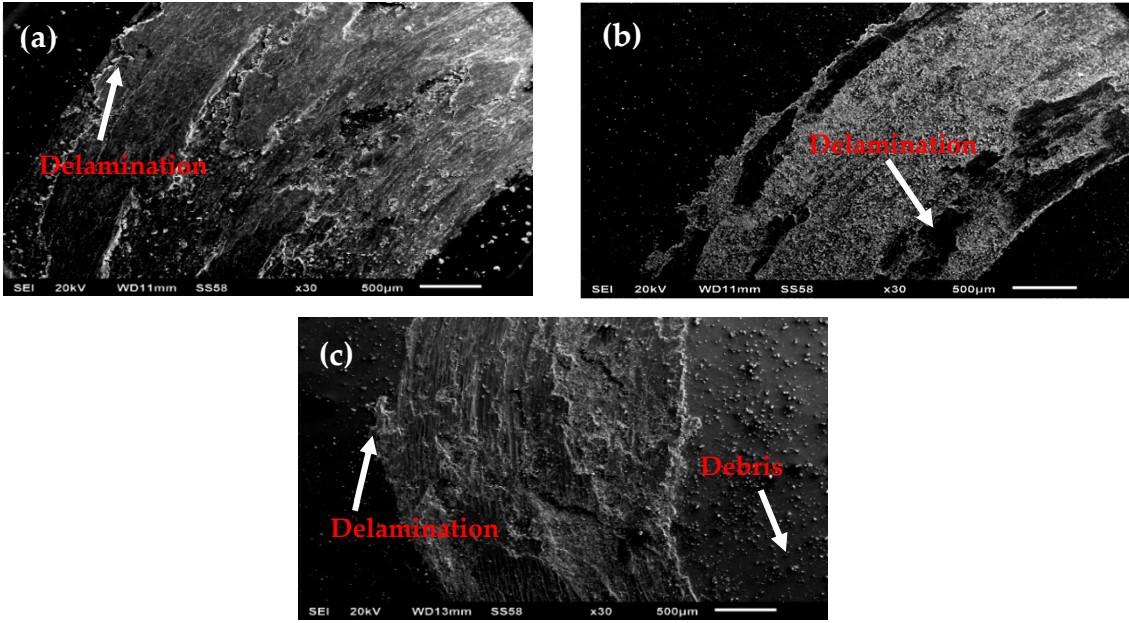

**Figure 9.** SEM micrographs at 30× of the wear tracks of samples: (**a**) A1S3; (**b**) A2S3; (**c**) commercial alloy C17530.

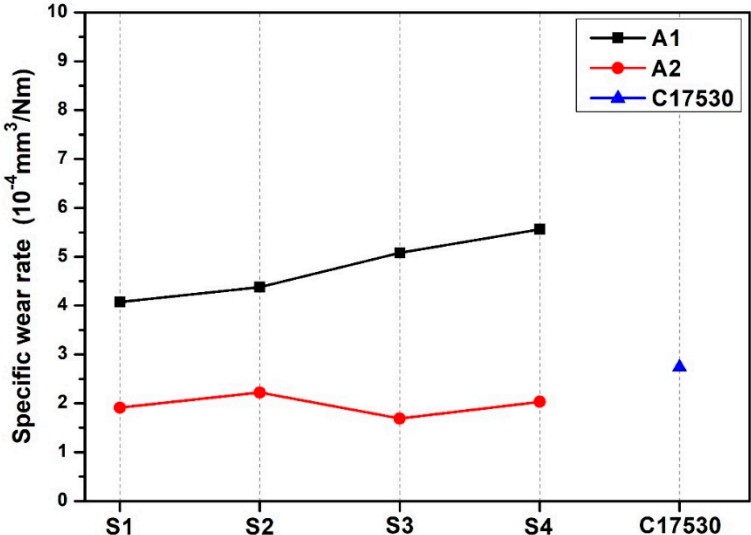

**Figure 10.** Wear rate variation for A1 and A2 alloys with different heat treatments on the samples S1, S2, S3 and S4 and C17530 alloy.

## 4. Conclusions

B and Nb additions on A1 and A2 alloys with a solution treatment and two aging were studiedin order to understand the effects on the microstructure and properties, conclusions are summarized as follows:

- There is an evolution of the microstructure due to B and Nb additions and heat treatments; however, this does not seem to have any effect on the SDAS.
- There is a change in grain size as heat treatments are applied; it decreases from 1500 to 900 μm for A1 alloy, and from 1700 to 1100 μm for A2 alloy.
- There are some precipitate phase growths in the grain boundary after conducting aging heat treatments: $B1Cr1$, $CoNi$, $Cu0.9Ni0.1$ and $B1.43Ni4.29Si2$ for A1 alloy, and $CrSi2$, $Nb6Ni16Si7$, $Cu3Fe17$ and $CoNi$ for A2 alloy.
- B and Nb additions with the heat treatments increased HRB hardness, with A1S3 having the highest result of $100.7 \pm 0.57$ HRB, which is an increment with the base alloy, and higher than the commercial alloy C17530 $85 \pm 0.57$ HRB.
- Specific wear rate was decreased, with B additions having the best result in the A1S1 condition with a specific wear rate of $4.07 \times 10^{-4}$ mm$^3$/Nm. With Nb additions, specific wear rate increased, with the A2S3 condition $1.69 \times 10^{-4}$ mm$^3$/Nm having the best result, while for the C17530 alloy, this was $2.74 \times 10^{-4}$ mm$^3$/Nm.

## 5. Patent

Register MX/a/2018/015771

**Author Contributions:** Conceptualization, D.A.A.-S. and M.A.L.H.-R.; Formal analysis, D.A.A.-S., A.J.-H. and M.A.L.H.-R.; Investigation, D.A.A.-S. and M.L.B.; Methodology, D.A.A.-S.; Project administration, A.J.-H. and M.A.L.H.-R.; Resources, A.J.-H.; Software, F.M.-O.; Supervision, A.J.-H.; Visualization, F.M.-O. and M.L.B.; Writing–original draft, D.A.A.-S.; Writing–review & editing, A.J.-H. and M.A.L.H.-R. All authors have read and agreed to the published version of the manuscript.

**Funding:** One of the authors, D.A. Avila Salgado, thanks CONACYT-México and Altea Casting S.A de C.V.

**Acknowledgments:** This work is supported by CONACYT and Altea Casting S.A de C.V

**Conflicts of Interest:** The authors declare that they have no conflict of interest.

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
