# Peer review of "Influence of B and Nb Additions and Heat Treatments on the Mechanical Properties of Cu-Ni-Co-Cr-Si Alloy for High Pressure Die Casting Applications"

_metals, doi:10.3390/met10050602_

Round 1

Reviewer 1 Report

The authors study the influence of heat treatment and B, Nb additions on the microstructure, hardness and wear resistance of Cu-Ni-Co-Cr-Si based alloys. This type of alloys applied for production of electronic materials. Authors used the optical microscopy, scanning electron microscopy and X-Ray diffraction techniques for analysis of microstructure and as well as the hardness and wear measurements for analysis of mechanical properties. The work is interesting and original, however the text of manuscript is written in a very complicated way. Often sentences are very long (up to 5 or 9 lines), which makes it difficult to understand them.

Grammatical errors are also common. In my opinion, the text should be rewritten completely.

I recommend publishing this manuscript after major corrections.

Abstract must be rewritten.

The second sentence of abstract is not completed.

The fourth sentence is not clear and complicated for understanding.

The sentence about hardness is also not clear for understanding. How can hardness increase and decrease from 80 to 100 HRB for the both alloy?

In my opinion, the last sentence also not correctly written, because the sizes of 6.31 and 4.44 mm3 refer to disc volume loss, not wear resistance.

It is not clear from the abstract what C17530 alloy is and what is its relation to the examined alloys in the work?

Introduction.

The sentences “Copper increases its performance when it is alloyed with different elements such as electric resistance” (line 35-36), and “Ti can refine the grain significantly by Cr and Cr3 phases and Si particles” are strange for me.

The sentences from 47 to 51 lines and from 56 to 65 lines are too long and should be divided into several sentences.

Authors added 1.3wt.% of Fe to one examined alloy, however is no information about the reason for this addition. What is effect of iron on properties of the examined alloy?

Materials and Methods

The authors wrote that “the material used in the present work was made of Cu, Si, Co, Cr, 99.99 wt% purity”. The question is: how were Fe, Ni, B and Nb added to the material?

Please check that the name of the scanning microscope is correct: MEB (?) JEOL JSM-6510LV.

The abbreviation SDAS means Secondary Dendrite Arm Spacing. Please correct it (line 101).

Please write units for parameters R, d and r.

Results and Discussion

The first sentence has 9 lines! Please divide it into several sentences!

Fig. 4 should show the microstructure images of the S1-S4 samples. However, the microstructure images of the S1 and S2 samples are missing. Please add them!

About the width of which peak is spoken by the authors in the second sentence of chapter 3.2 (line 167)?

According to the Fig.7a, the phase Cu0.9Si0.1 belongs to the A1 alloy (not to the A2 alloy as it is written on the line 173).

The present of the CuNi phase was stated in the A2 alloy (line 173). However where are the peaks of this phase on the XRD patterns, which are presented in the Fig.7b?  

What is mean “abrasive mechanism was higher”? (line 221)

Please rewrite the sentence “ In figure 10 is…” (lines 222-224) for better understanding.

The authors wrote that “it was observed that diffraction angle for Cu was moved as crystal parameters were modified due to the impurities in the matrix”. The peaks were shifted to the left or right side and as a result of increase or decrease of lattice parameters?

Conclusions

Please  check that the phase names are correct in conclusion 3.
Are the authors sure that wear resistance has units mm3? (conclusion 6).

References.

Write references in the same way. Pay attention to the Ref. 1, 20, 22-25, 29.

Author Response

Greetings dear reviewer, in this annex the reviews and the article with all the corrections. 

Reviewer 2 Report

Presented work is valuable, but have minor shortcomings. 

What is the recommendation for the reader? What is it better to add to the alloy, B or Nb? What heat treatment gives the best results? Please clarify this in the conclusions.

Figure 2 is on two pages.

The legend of Figure 4 is on the other page.

Line 227 there is a too space, "...matrix,   so..."

Author Response

(The authors gave the same response as above.)

Reviewer 3 Report

The authors compare the effect of B and Nb alloying additions on the microstructure and mechanical properties of a Cu-Ni-Co-Cr-Si alloy, intended for application in HPLC die plungers.

General comments:

  • Experimental design

Did the authors examine, or is there previous work, on alloys of the *identical* base composition, but without either B or Nb; i.e. a control sample to properly indicate the effectiveness of the critical alloying elements? How can the authors clearly distinguish the effect of B and/or Nb in the absence of such a base sample? What was the rationale for the amount of B or Nb chosen?

  • Grain size and dendrite spacing measurements.

The authors point out the microstucture in the castings is highly position-dependent, with a transition between a columnar and equiaxed grain morphology. This complicates the measurement of grain size and SDAS; where were the measurements made in each sample? Were the measurements made at the same location in each case? How did the spacial variation in grain size compare with the change due to alloy composition.

  • Xray analysis

Sampling location may also play a role in the Xray analysis. The change in ratio of the Cu peaks strongly suggests differences in the crystallographic texture of the samples. This could arise from differences in the microstructure due to position as well as to heat treatment. This is not a problem for qualitative identification of the phases, but should be resolved to make inferences about the peak intensity and grain size, as the authors do.

Specific queries;

Introduction

Page 1 Line 45-46: (regarding Berylliosis) "Be gas ... Be fumes or particles" What is the form of Be responsible for Berylliosis? Would casting Be result in a significant vapor pressure of the metal?

Materials and methods

*  What were the actual durations of the heat-treatments? Table 2 gives time ranges (e.g. S4, 30-180 minutes), but it is not indicated what treatment times the hardness, optical microscopy or Xrd results relate to. Significant changes in hardness and microstructure would be expected to occur over such a broad time range. Were hardness (or other) results obtained at multiple ageing times?

Results and discussion

Page 3 line 99  The author state that optical and scanning electron microscopies were used for sample characterisation, but results are only presented for optical microscopy. What results were obtained from SEM?

(It should be possible to detect the grain boundary phases referred to by the authors, and possibly determine a composition from energy dispersive x-ray spectroscopy.)

Figure 4 a) and b) are missing, possibly cut off at a page break.

Page 7 Line 172 Xray analysis

"Some precipitated (sic.) were found in the grain boundary after ageing heat treatments, B1Cr1, CoNi, Cu0.9Ni0.1 and B1.43Ni4.29Si2 for A1 alloy, while CuNi, Cu0.9Si0.1 for A2 alloy."

  • The Xray results are not spatially-resolved, therefore it is not possible to state where a secondary phase is located from the spectra.
  • How were the secondary phases identified? Could the authors provide references for the secondary phases identified in the spectra?
  • Peaks (at least for the major phase) should be labelled with hkl index.

Summary:

The work is of interest, but I do not believe publishable in its current form. The absence of a base sample makes is impossible to isolate the effect of a single alloying element (B,Nb).

Additionally, the ambiguity about the heat treatment times which the optical, hardness and xrd results relate would make it impractical for other workers to reproduce the experiments.

I would encourage the authors to address these issues and resubmit the article.

Author Response

(The authors gave the same response as above.)

Round 2

Reviewer 1 Report

Dear Editor,

The article is corrected and the author's responses are satisfying. I have no questions and recommend publishing this paper after some gramma corrections. For example, peaksin - line 178, treatmentsfor - line 186,  alloyswere - line 259, CoNifor - line 266; two dots at the end of the sentence can be seen - line 21, or an unnecessary dot in the middle of the sentence – line 28 and so on.

Reviewer 3 Report

The authors have made a thorough revision, which address some concerns raised with the previous version. I would suggest only a check in proof for any typographical or similar errors.